# Tenofovir Alafenamide Rescues Renal Tubules in Patients with Chronic Hepatitis B

**DOI:** 10.3390/life11030263

**Published:** 2021-03-23

**Authors:** Tomoya Sano, Takumi Kawaguchi, Tatsuya Ide, Keisuke Amano, Reiichiro Kuwahara, Teruko Arinaga-Hino, Takuji Torimura

**Affiliations:** Division of Gastroenterology, Department of Medicine, Kurume University School of Medicine, Kurume, Fukuoka 830-0011, Japan; takumi@med.kurume-u.ac.jp (T.K.); ide@med.kurume-u.ac.jp (T.I.); amano_keisuke@kurume-u.ac.jp (K.A.); ray@med.kurume-u.ac.jp (R.K.); terukoh@med.kurume-u.ac.jp (T.A.-H.); tori@med.kurume-u.ac.jp (T.T.)

**Keywords:** adefovir dipivoxil (ADV), Fanconi syndrome, hepatitis B virus (HBV), renal tubular dysfunction, tenofovir alafenamide (TAF), tenofovir disoproxil fumarate (TDF), β2-microglobulin

## Abstract

Nucles(t)ide analogs (NAs) are effective for chronic hepatitis B (CHB). NAs suppress hepatic decompensation and hepatocarcinogenesis, leading to a dramatic improvement of the natural course of patients with CHB. However, renal dysfunction is becoming an important issue for the management of CHB. Renal dysfunction develops in patients with the long-term treatment of NAs including adefovir dipivoxil and tenofovir disoproxil fumarate. Recently, several studies have reported that the newly approved tenofovir alafenamide (TAF) has a safe profile for the kidney due to greater plasma stability. In this mini-review, we discuss the effectiveness of switching to TAF for NAs-related renal tubular dysfunction in patients with CHB.

## 1. Introduction

Hepatitis B virus (HBV) infection is the most common chronic hepatitis viral infection worldwide. HBV infects approximately 2 billion individuals, of which greater than 350 million are chronic HBV carriers [1]. Although chronic hepatitis B (CHB) increases the risk of hepatic decompensation and hepatocellular carcinoma [2,3], the development of nucleos(t)ide analogs (NAs) has dramatically improved the natural course of patients with CHB in the last two decades [4].

NAs for HBV are classified according to their chemical structures. The 1st generation NAs include lamivudine (LAM) and entecavir (ETV) and the second generation NAs include adefovir dipivoxil (ADV) and tenofovir disoproxil fumarate (TDF) [5]. A randomized controlled study demonstrated that treatment with LAM significantly suppressed hepatocarcinogenesis compared to the placebo group with a hazard ratio of 0.49 [6]. In addition, long-term ETV treatment has been reported to inhibit hepatocarcinogenesis compared to patients with non-treated CHB with a hazard ratio of 0.37 [7]. NAs are generally safe and comparatively free of severe side effects [8]. However, renal tubular dysfunction has been known to develop in CHB patients, treated with NAs, including ADV and TDF [9]. Renal impairment and hypophosphatemia have been observed in 10.5% and 26.7% of patients with long-term administration of ADV, even low-dose (10 mg/day) of ADV, respectively [10]. Moreover, six months after treatment with TDF, 59.5% of patients showed elevated urinary β2-microglobulin (U-BMG), which is a sensitive marker for renal tubular dysfunction [11]. Furthermore, bone metabolism abnormalities and subsequent bone fracture have been reported to develop in patients with the long-term use of ADV/TDF [12,13,14,15]. As the life expectancy of HBV-infected individuals has increased, the long-term adverse effects of antiviral therapies have increasingly emerged [16]. Therefore, management of renal tubular dysfunction is becoming an important issue for the management of CHB patients treated with ADV/TDF.

## 2. Mechanisms for the ADV/TDF-Related Renal Tubular Dysfunction and Fanconi Syndrome

The detail of mechanisms for ADV/TDF-related renal tubular dysfunction remains unclear. However, mitochondrial dysfunction of the proximal tubule cells is presumed to be a possible mechanism of renal dysfunction and the onset of Fanconi syndrome [17]. In addition, multi-drug resistance-associated protein 2 and 4 are involved in the excretion of ADV from the renal tubule and the genetic polymorphisms of these transporters have been reported as risk factors for ADV-related renal impairment [18,19]. In Japanese patients with human immunodeficiency virus (HIV)-1 infection, single nucleotide polymorphisms in adenosine triphosphate–binding cassette C2 has been reported to be associated with TDF-induced renal tubular dysfunction [20].

Fanconi syndrome is a disease that causes disorders of glucose, amino acid, phosphorus, and bicarbonate reabsorption in the proximal tubule of the kidney, and is often associated with osteomalacia. Consequently, it results in serious complications, such as multiple bone fractures [21]. In fact, more than 150 cases of hypophosphatemia osteomalacia or Fanconi syndrome have been reported with long-term HBV treatment by ADV [22,23]. Risk factors for ADV-induced Fanconi syndrome have been reported. Male sex, age ≥ 40 years, decreased eGFR at the start of ADV treatment, hypertension, diabetes, cirrhosis, East Asian ethnicity, low body mass index, treatment with ADV for more than 24 months, residence in rural areas, and prior use of nephrotoxic drugs as risk factors for ADV-induced Fanconi syndrome in CHB patients [22,23,24]. TDF also has a risk for Fanconi syndrome in patients with HBV [25], although the risk factor remains unclear.

## 3. Efficacy and Safety of TAF

In 2017, a third generation NA, tenofovir alafenamide (TAF), has been approved in Japan [26,27,28,29]. A recent network meta-analysis of 42 randomized controlled trials reported that TAF, along with TDF, is a recommended medication for a virologic response and ALT normalization for the treatment of CHB [30]. In addition, no TAF-resistant HBV has been reported in 132 patients treated for 96 weeks [29]. No TAF-resistant HBV has been identified up to now in either naïve or treatment-experienced subjects [31].

TAF is a phosphonamidate prodrug of tenofovir (TFV) that shares the same intracellular active metabolite, TFV diphosphate, which is effective against HBV [32,33]. A feature of TAF is greater plasma stability, resulting in more efficient uptake by hepatocytes at lower plasma concentrations than TDF. Therefore, the circulating concentration of TFV is 90% lower in TAF than in TDF [34] and this difference is thought to contribute to the better safety profile of TAF compared with TDF, particularly for renal tubular dysfunction and bone metabolism [26,27,28]. In international phase III trials, a decline of eGFR was significantly inhibited in the TAF group than the TDF group at week 96 (TAF median −1.2 mL/min vs. TDF −4.8 mL/min). Moreover, at week 144, a decline of eGFR was also significantly inhibited in the TAF group than the TDF group (TAF median −1.2 mL/min vs. TDF −6 mL/min) [28]. Urine levels of retinol-binding protein/Cr and BMG/Cr, renal tubular markers, were also significantly lower in the TAF group than the TDF group at week 96 and week 144 [28]. Phase III non-inferiority studies showed that a decline of bone mineral density (BMD) concentration in hip and spine was inhibited in the TAF group than the TDF group (Hip −0.33% vs. −2.51%; Spine −0.75% vs. −2.57%) at 144 weeks of treatment [26,27,28].

The effect of ETV on renal function was not significantly different from the untreated patients [35] and ETV is thought to have a small effect on renal function. However, several recent studies have reported the effectiveness of switching from ETV to TAF in reducing serum HBs antigen levels. Hagiwara, et al. reported that it was particularly prominent in patients with serum HBs antigen levels < 800 IU/mL after switching from ETV to TAF [36]. Uchida et al. found that the degree of reduction in serum HBs antigen levels after switching from ETV to TAF was significantly higher, particularly in patients with cirrhosis, genotype B HBV infection, and serum hepatitis B core related antigen levels <3.0 log U/mL [37].

In relation to HBV-infected liver transplant recipients, TAF was associated with high antiviral efficacy to prevent reactivation, as well as less decline in renal function when compared to other NAs [38,39]. Furthermore, TAF has some other strengths. TAF has been reported to be safe for both mothers and infants [40]. The transmission rate of TAF from mother to infant was 0% and no infant has birth defect [40]. Moreover, TAF improved adherence due to the convenience of administration timing compared to ETV, as TAF absorption is unaffected by food intake [41]. In addition, TAF has better cost-effectiveness compared with both TDF and ETV [42].

## 4. Adverse Events of TAF

The incidence for adverse events (AEs) of TAF was 14.2%. The most commonly reported AEs were nausea (2.1%), tiredness (1.4%), and headache (1.4%) [26,27]. In a recent review, the following AEs have been reported: nasopharyngitis (10.2%), occult blood stool (7.1%), elevated ALT (8.6%), elevated LDL cholesterol (4.0%), urine erythrocytes (7.8%), urine glucose (4.9%), and increased creatine kinase (2.6%) [43]. Although TAF is considered a well-tolerated treatment, 1% of patients discontinued treatment with TAF due to AEs in international phase III trials [26,27]. Therefore, we have to be cautious about TAF-related AEs. However, the number of patients treated with TAF still is too small in comparison with other NAs, and further studies are required.

We also have to pay attention to body weight in HBV patients treated with TAF. Recent studies have shown that switching from a TDF to a TAF regimen was associate with an increase in weight gain after long-term follow-up in patients with HIV [44,45,46,47]. The reason for this weight gain is unknown. However, the TAF-related increase in body weight has been reported not only in HIV patients but also in patients with HBV [48].

## 5. Effects of Switching to TAF on NAs-Related Renal Tubular Dysfunction

In clinical practice, switching to TAF is an important issue in CHB patients with NAs-related renal tubular dysfunction. So far, six studies have been reported on the effects of switching from ADV/TDF to TAF on renal function in patients with HBV (Table 1). Only one study reported exacerbation creatinine clearance (CCr) levels [48]. The reason for the exacerbation remains unclear. However, the majority of the enrolled subjects showed normal renal function at the baseline and, therefore, enrolled subjects were not suitable for the evaluation of the effect of TAF on NAs-related renal tubular dysfunction [48]. On the other hand, the other five studies have demonstrated that switching from ADV/TDF to TAF improved renal function [49,50,51,52,53]. Three prospective single-arm open-label studies demonstrated that beneficial effects on renal functions and BMD were observed in the patients switched to TAF than in patients treated with TDF [48,51,52]. Furthermore, Lampertico P. et al. performed a randomized controlled trial with 488 CHB patients treated with TDF. They reported an increase in CCr was observed in the patients switched to TAF (n = 243) than in patients treated with TDF (n = 254) (median change 0.94 mL/min [IQR −4.47 to 6.24] versus −2.74 mL/min [−7.89 to 1.88]) at week 48 [49]. Lee BT et al. reported that the proximal tubular function improved at the long-term of 72 weeks compared to baseline [48]. In addition, Ogawa E. et al. reported that switching to TAF improves renal dysfunction associated with various NA combinations (LAM/ETV and ADV/TDF) [50]. We also reported that switching from ADV/TDF to TAF improves U-BMG/Cr ratio and bone specific alkaline phosphatase even in CHB patients with long-term treatment of ADV (9.8 ± 3.0 years) [53]. Therefore, switching to TAF is an important therapeutic strategy for CHB patients with NAs-related renal tubular dysfunction.

## 6. Clinical Profile for the ADV/TDF-Related Renal Tubular Dysfunction

Furthermore, we performed a decision-tree analysis to reveal patient characteristics associated with ADV/TDF-related renal tubular dysfunction, which was defined as >300 μg/g Cre of U-BMG/Cr [54]. A decision-tree algorithm is a data-mining technique that reveals a series of classification rules by identifying priorities. It has been used to identify the profiles associated with the progression of chronic kidney disease [55]. The decision-tree analysis showed that age was the initial classifier for the ADV/TDF-related renal tubular dysfunction. The prevalence of renal tubular dysfunction was 46.2% in patients under 61 years of age (Figure 1). On the other hand, in patients with ≥61 years of age, the prevalence of renal tubular dysfunction was 90.0% (Figure 1). It remains unclear why age was the most important factor for ADV/TDF-related renal tubular dysfunction. However, a possible explanation is that the renal plasma flow is reduced in elderly subjects [56]. Furthermore, age-associated telomere shortening is reported to be a possible factor for increased tubular injury and limited regenerative response after renal injury [57]. Di Perri reported that TDF should be avoided in elderly patients considering its effect on renal function and bone metabolism over a long period, supporting the results of our study [58]. In addition, the previous reports revealed that long-term TDF treatment can cause clinically significant nephrotoxicity, especially in patients over 60 years old and with baseline renal impairment [59]. More importantly, EASL guidelines proposed switching CHB patients older than 60 years or with bone or renal disease to TAF or ETV to overcome the safety limitations of TDF [60]. The reason for the switching to TAF is a high prevalence of complications. In CHB patients treated with TDF (n = 565), the high prevalence of type 2 diabetes (15%) and hypertension (50%) was seen in subjects over 60 years of age [60]. Moreover, a high prevalence of renal dysfunction (32%) and hypophosphatemia (25%) were noted in those over 60 years of age [60]. Accordingly, approximately two-thirds of patients receiving long-term TDF are a candidate for an ETV or TAF switch and EASL recommendations stated that TAF may be the most appropriate therapeutic option for most of CHB patients given the previous exposure to NAs [60]. Our study has several limitations including small sample size. However, our findings, along with previous studies suggest that age is an important factor to consider switching TAF from ADV/TDF in patients with CHB.

## 7. Conclusions

In this mini-review, we focused on NAs-related renal tubular dysfunction, which is becoming an important issue in the era of long-term NA treatment for CHB. We reviewed previous studies and proposed that switching to TAF is an important therapeutic strategy for CHB patients with NAs-related renal tubular dysfunction, in particular, in patients ≥61 years of age. We conclude this short review with the following four bullet points:Management of renal tubular dysfunction is becoming an important issue for the management of CHB patients treated with ADV/TDF.Switching to TAF is an important therapeutic strategy for CHB patients with NAs-related renal tubular dysfunction.Age, in particular ≥61 years old, is an important factor to consider switching TAF from ADV/TDF in patients with CHB.The number of patients treated with TAF still is too small in comparison to other NAs. It is required to accumulate evidence about TAF-related AEs.

## Figures and Tables

**Figure 1 life-11-00263-f001:**
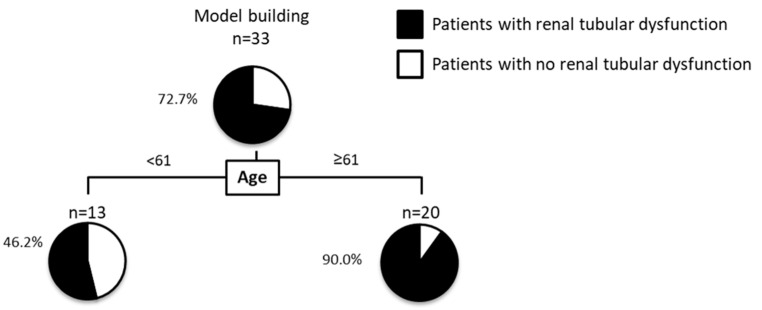
Decision-tree algorithm for renal tubular dysfunction. Renal tubular dysfunction was defined as >300 μg/g Cre of U-BMG/Cr. The pie graphs indicate the proportion of patients with renal tubular dysfunction (black) and patients with no renal tubular dysfunction (white).

**Table 1 life-11-00263-t001:** Effects of switching to TAF from other NAs including ADV/TDF on renal function.

Author Reference	Study Design	n	Intervention	Assessment Point after Intervention	Outcome: Renal Function	Outcome: Bone Metabolism	Reference
Lampertico P. et al. Lancet Gatroenterol Hepatol. 2020	Phase III RCT	488	TDF→TAF	48 weeks	Improvement of CCr	Improvement of BMD	[49]
Ogawa E. et al. Liver Int. 2020	Multicenter retrospective cohort study	122	NA combination * →TAF	48 weeks	Improvement of eGFR and U-BMG/Cr	Improvement of serum P	[50]
Fong TL. et al. J Viral Hepat. 2019	Prospective single-arm open-label study	75	TDF→TAF	24 weeks	Improvement of U-BMG/Cr and U-RBP/Cr	Improvement of BMD	[51]
Lee BT. et al. JGH Open. 2020	Prospective single-arm open-label study	61	TDF→TAF	72 weeks	Improvement of U-BMG/Cr and U-RBP/Cr, Exacerbation of CCr	Improvement of BMD	[48]
Kaneko S. et al. J Gasrienterol Hepatol. 2019	Prospective single-arm open-label study	36	TDF→TAF	24 weeks	Improvement of eGFR and U-BMG/Cr	Not applicable	[52]
Sano T. et al. Biomed Rep. 2021	Retrospective observational study	33	ADV/TDF→TAF	24 weeks	Improvement of U-BMG/Cr	Improvement of ALP and BAP	[53]

Note. * The NA combination includes LAM/ETV and ADV/TDF treatments. Abbreviations: TAF, tenofovir alafenamide; NA, nucleos(t)ide analog; ADV, adefovir dipivoxil; TDF, tenofovir disoproxil fumarate; RCT, Randomized Controlled Trial; CCr, creatinine clearance; BMD, bone mineral density; eGFR, estimated glomerular filtration rate; U-BMG/Cr, urine β2-microglobulin-creatinine ratio; P, phosphorus; U-RBP/Cr, urine retinol-binding protein-creatinine ratio; ALP, alkaline phosphatase; BAP, bone specific alkaline phosphatase; LAM, lamivudine; ETV, entecavir.

## Data Availability

The data presented in this study are available on request from the corresponding author. The data are not publicly available due to ethical reason.

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
