# Peer review of "Tenofovir Alafenamide Rescues Renal Tubules in Patients with Chronic Hepatitis B"

_life, 2021, doi:10.3390/life11030263_

Round 1

Reviewer 1 Report

This is an excellent, well-written review paper that briefly describes the perspectives of TAF for coping with NAs-related renal tubular dysfunction in patients with CHB. Introduction parti provides sufficient background for the general audience. Next, the authors discuss the mechanisms of ADV/TDF-induced renal tubular dysfunction as well as risk factors. Safety and efficacy of TAF and reports about the effects of the switch to TAF therapy are summarized further.

To conclude, this is a concise, well-written and timely manuscript that could be interesting to general practicioners, gastroenterologists and hepatologists.

One major comment that I have is to ask the authors to broaden the Conclusion section and provide the bullet points with the key findings, summary and brief final conclusions and perspectives.

Author Response

To REVIEWER 1

Thank you very much for your letter regarding our manuscript (life-1130315). We appreciate your time and effort in reviewing our manuscript. We also appreciate your comments, which have helped us to improve our manuscript. In line with your comments, please find below our response.

Comment 1: Broaden the Conclusion section and provide the bullet points with the key findings, summary and brief final conclusions and perspectives.

Answer: We appreciate your comment. As you suggested, we have broadened the Conclusion section and provided the bullet points with the key findings, summary, brief final conclusions, and perspectives as below (line 199-207).

We conclude this mini-review with the following four bullet points:

  • Management of renal tubular dysfunction is becoming an important issue for the management of CHB patients treated with ADV/TDF. 
  • Switching to TAF is an important therapeutic strategy for CHB patients with NAs-related renal tubular dysfunction. 
  • Age, in particular ³61 years old, is an important factor to consider switching TAF from ADV/TDF in patients with CHB. 
  • The number of patients treated with TAF still is too small in comparison to other NAs. It is required to accumulate evidence about TAF-related AEs. 

Again, we appreciate your comments, which have helped us to improve our manuscript.

Reviewer 2 Report

At more than 1.5million deaths per year worldwide, viral hepatitis B and C constitutute of the  most common causes of death. Despite the availibilty of a vaccination against the hepatitis B virus ( HBV), the prevalence of chronic HBV infection has reduced only slightly over the past three decades, from 4,2% in 1990 to 3,7% in 2005. In addition to this possibility of prevention from acute infections, the use of nucleos(t)ide analogues ( NAS) is a decisive and promising measure for controlling chronic HBV infection and countering the development of liver cirrhosis with all complications, especially hepatocellular carcinoma ( HCC). Due to the integration  of the "circular covalently closed" DNA ( ccc DNA) into the genome of the hepatocyte, curing the HBV infection does not seem possible at present. However, the latest studies encouraging progress in this regard. At present, besides entecavir (ETV), adefovir dipivoxil (ADV) and the various forms of tenofovir (TFV; adefovir disoproxil fumarate TDF; and tenofovir alafenamide fumarate, TAF) are most effective treatment options for chronic hepatitis B. With these medications, it is often possible to achieve unlimited control of viral replication ( HBV-DNA < 2000 IU/l)  and lack of inflammatory activity ( normal transaminases).

All measures which aim to improve therapeuticc effectivenes in hepatitis B are highly relevant. The paper present comparing the incidence of side effects, especially  renal side effects, of TDF and TAF is highly novel.

Following a brief introduction to the epidemiology of the HBV infection, the authors concisely describe the mechanisms of ADV/TDF-induced renal tubular damage and Fanconi syndrome. As a prodrug of TFV, the circulating concentration of TAF is up to 90% lower than in TDF. The authors consider this difference to be the decisive advantage of TAF regarding the side-effect profile, in particular with regard to the kidneys and bones. The antiviral efficacy of TAF is comparable to ETV, and in some even better. The transmission rate from mother to child is 0%. Malformations did not occur either. Weight gain with TAF is unclear. In most cases, the switch from ADV/ TDF to TAF is associated with an improvement in kidney function in bone metabolism. Given that age ( > 60 years) is a critical prognostic factor for renal tubular dysfunction, patients of this age  or with diseases of the kidneys or also bones should swithched from ADV / TDF to TAF or ETV.

I a short and concise form, the paper gives the clinician an effective overview of the mechanism and side effects of TAF as well as the advantages over ADV and TDF. The statements are factual, written in legible English, and are well substantiated by suitable collection of literature ( 66 citations).

I recommend the acceptance of the paper in present form.

Author Response

To REVIEWER 2

Thank you very much for your letter regarding our manuscript (life-1130315). We appreciate your time and effort in reviewing our manuscript. We appreciate that you understand the importance of our manuscript. In addition, we are very glad to hear that you could recommend the acceptance of the paper in the present form. Again, we deeply appreciate your time and effort in reviewing our manuscript.
